# Faith in Humanity: Religious Charitable Organizations Solidarity towards Migrants in the United Arab Emirates

Wafa Barhoumi Hamdi [1,*], Semiyu Adejare Aderibigbe [1,2,*], Mesut Idriz [1] and Mouza Mohamed Alghfeli [1]

1   College of Arts, Humanities and Social Sciences, University of Sharjah, Sharjah P.O. Box 27272,
    United Arab Emirates; m.idriz@sharjah.ac.ae (M.I.); u19106079@sharjah.ac.ae (M.M.A.)
2   Institute of Leadership in Higher Education, University of Sharjah, Sharjah P.O. Box 27272,
    United Arab Emirates
*   Correspondence: whamdi@sharjah.ac.ae (W.B.H.); saderibigbe@sharjah.ac.ae (S.A.A.)

**Abstract:** The plight of migrants and the need to foster their integration into diverse societies are of concern to global communities, governments, and charitable organizations. This study explored the roles of philanthropic organizations in the United Arab Emirates (UAE) in demonstrating solidarity towards migrants, exploring the multifaceted dimensions of compassion and support rooted in diverse religious principles. The study adopted a qualitative approach guided by the interpretivist paradigm when collecting and analyzing data. From its findings, the study highlights the integral role they play in addressing the complex needs of migrant communities. For instance, they provide food and financial support in clearing hospitals, tuition, and rent bills. In addition, the findings show that the organizations identify migrants in need through their open-door policy, encouraging those in need of support to contact them directly, with collaboration also being essential for the effectiveness of their services. By affirming the positive contributions of religious charities, this study underscores their role in promoting the well-being and social cohesion of migrants, ultimately exemplifying a shared commitment to humanitarian values and reinstating faith in the collective capacity for compassion and solidarity.

**Keywords:** religion; solidarity; UAE; migrants; charitable organizations

## 1. Introduction

In an era of increasing global interconnection and diversity, the movement of migrants and refugees across borders has emerged as a critical global concern (García-Cid et al. 2020). Often fleeing conflict, persecution, or economic adversity, these individuals and families seek safety and a better life. The integration process becomes pivotal as they settle in new lands, with various factors contributing to their success, including respect for diversity and tolerance (United Nations 2018). Religious solidarity grounded in spiritual philosophies and principles is one powerful force in this equation requiring more attention.

As an Arab Gulf nation, the United Arab Emirates (UAE) admonished its citizens and residents to unite in responsibility, destiny, knowledge, and prosperity as critical pillars of its Vision 2021 plan (https://uaecabinet.ae/en/uae-vision, accessed on 2 September 2023). It also reinforced its commitment to equality and social justice for its growing and diverse populace, including citizens and residents, by launching the National Human Rights Authority in 2020 (https://www.uae-embassy.org/discover-uae/society/human-rights, accessed on 2 September 2023). In nurturing and fostering diversity and religious tolerance, the UAE government initiated measures, including the Year of Tolerance and the Abrahamic Family House for interfaith dialogue (Aderibigbe et al. 2023). Additionally, the UAE is involved in intercultural solidarity by backing donor and charitable agencies undertaking humanitarian works within and outside the UAE, including religion-based solidarity initiatives aiding migrants and refugees (Cochrane 2021; UAE Foreign Aid 2021. https://www.mofa.gov.ae/en/The-Ministry/UAE-International-Development-

Cooperation/Annual-Foreign-Aid-Report, accessed on 17 October 2023). Not surprisingly, it has recently gained credence as one of the most generous humanitarian aid donors (Gökalp 2022, p. 2).

The commitment of the UAE to diversity has paved the way for a society that values and respects different faiths, creating an environment where migrants feel welcomed and appreciated for their religious beliefs. However, the need to strengthen the process of supporting migrants through charitable organizations requires meticulous plans and studies, exploring their motivation and strategies to enhance their operation. Cochrane (2021) argued that aid-related issues and portfolios in the UAE are under-researched, and efforts in this regard are essential.

In our view, engaging in regular research endeavors becomes necessary to understand issues and enhance charitable organizations' operations as they keep evolving and migrants' needs keep changing. Such moves will also provide insights into the approaches used to help immigrants by recommending how individuals and other organizations can complement the charitable organizations' efforts. Thus, we explored the roles of charitable organizations guided by religious principles in providing relief aid and fostering social cohesion within the UAE. We contend that the study contributes to the international literature on the role of religious organizations in supporting the integration process for migrants within different cultural contexts, drawing on the UAE experience.

## 2. Review of the Related Literature

### 2.1. Religious Values and Charitable Organizations

The intersection of religious and charitable organizations and their role in providing solidarity toward migrants has been a topic of considerable scholarly interest. As migration continues to be a global phenomenon, religious institutions have emerged as key players in addressing the complex challenges faced by migrants, offering both material support and spiritual guidance. Numerous studies highlight the foundational role of religious beliefs in motivating charitable acts toward migrants. Scholars like Zanfrini (2020) emphasize the intrinsic connection between religious teachings, compassion, and the imperative to assist those in need.

Religious values often underpin the ethos of charity, establishing a moral obligation to extend support and solidarity to migrants within communities. Research by Johnson (2015) delves into the institutional frameworks of religious organizations involved in migrant solidarity. These institutions often have well-established programs that aid migrants, including shelters, legal aid services, and community integration initiatives. The study explores how the organizational structures of religious entities influence the efficacy and reach of their solidarity efforts. Interfaith collaborations have gained prominence in the context of migrant solidarity. The work of Khafagy (2020) demonstrates how partnerships between religious groups of different denominations foster a more comprehensive and inclusive approach to supporting migrants. These collaborations enhance the impact of religious charitable organizations by pooling resources, expertise, and networks.

While religious organizations play a pivotal role in migrant solidarity, research by Gomez et al. (2020) sheds light on the challenges and ethical considerations involved. Issues such as proselytization, cultural sensitivity, and potential conflicts with secular humanitarian principles are explored. The study emphasizes the importance of addressing these complexities to ensure respectful and inclusive solidarity efforts. A growing body of literature, including the work of Xu and Zheng (2023), focuses on the impact of religious charitable organizations on the social integration of migrants. Beyond immediate material assistance, these organizations often provide a sense of community and belonging, facilitating the integration of migrants into their new environments. The study examines the long-term effects of religious-based solidarity on the well-being and social cohesion of migrant populations.

Common religious philosophies informing principles to promote tolerance, diversity, and cohesion among diverse populace include love for neighbors, compassion toward

others, and acting in a welcoming manner. In Christianity, the central teaching of "love thy neighbor" from the Bible, exemplified in the Good Samaritan parable (Luke 10: 25–37), underscores the importance of showing compassion and hospitality to those in need. Pope Francis, in his encyclical "Laudato Si'" (2015), emphasizes the interconnectedness of humanity, urging the practice of love and care for one another in all circumstances. Islam stresses the significance of compassion and charity towards others, including migrants and refugees. The Quran, in Surah Al-Baqarah (2: 267), emphasizes the duty of sharing one's blessings with those less fortunate. The following hadith of the Prophet Muhammad (PBUH) outlines three deeds that continue after death: "When a man dies, his good deeds come to an end, except three: Ongoing charity, beneficial knowledge, and a righteous child who will pray for him." (Hadith: Al-Bukhari). This duty extends to migrants, reflecting the Islamic principle of caring for those in need. Buddhism centers on compassion (karuna) as a core value, promoting the idea of extending compassion to all sentient beings. The Dalai Lama, in "The Art of Happiness" (Tenzin 1998), discusses the importance of compassion, while Buddhist teachings underscore non-harm (ahimsa) and the interconnectedness of all beings, guiding the treatment of migrants with kindness and respect.

*2.2. Fostering Religious Solidarity for Migrant Integration: Strategies and Insights*

In the face of escalating global migration, religious solidarity is a crucial and intricate undertaking involving diverse stakeholders, including religious institutions, leaders, and communities. Solidarity, as a unifying force, has the potential to cultivate empathy, understanding, and cooperation among people of different faiths, thereby contributing to more inclusive and harmonious societies. Here are some key strategies and insights:

Interfaith dialogues and collaboration: Interfaith dialogues allow religious leaders and followers from diverse traditions to converge, share experiences, and discuss shared concerns. These dialogues play a pivotal role in dispelling misconceptions, fostering mutual respect, and enhancing understanding (Baderin 2013), leading to harmonious engagement, empathy, and solidarity with migrants. Interfaith efforts can lead to greater understanding and empathy (Parliament of the World's Religions 2023).

Humanitarian assistance and charity: Religious organizations often engage in charitable activities to address the immediate needs of migrants while concurrently building bridges of solidarity, such as providing food, shelter, medical care, and legal assistance. They do so in line with their religious teachings and values, emphasizing compassion and helping those in need. The extant literature indicates that many religious traditions emphasize assisting those in need regardless of differences (Aderibigbe et al. 2023; Haynes 2011). The Pew Research Center also reports that religious groups, particularly Christian and Muslim organizations, have provided humanitarian aid and support to refugees and migrants globally (Connor 2012).

Education, advocacy, and community integration: Many religious communities educate and advocate for the rights and dignity of migrants. They educate people about the plights of immigrants, raise awareness about immigration issues, lobby for policy changes, and work to address the root causes of migration. In the United States, various religious groups, including the U.S. Conference of Catholic Bishops and the National Council of Churches, have consistently advocated for comprehensive immigration reform and supported policies that promote the rights of undocumented immigrants (U.S. Conference of Catholic Bishops 2021). Religious communities also facilitate cultural exchange and integration by organizing events that bring together migrants and host communities. These events promote understanding and solidarity. The Church of England, for instance, has organized initiatives to promote cultural exchange and integration between migrant communities and local congregations, fostering greater social cohesion.

*2.3. Migration Patterns and Charitable Organizations in the UAE*

Since its creation in 1971, the UAE has witnessed a significant influx of expatriates from all over the world with Asians accounting for the largest of the diverse population. This

migration pattern was primarily driven by economic opportunities and the development of critical sectors such as finance, real estate, and tourism (Froilan and Tsourapas 2021). Essentially, one of the key factors contributing to migration patterns in the UAE is the pursuit of employment opportunities. The country's robust economy and ambitious development projects, including real estate and tourism landmarks, have attracted a diverse pool of skilled and unskilled workers from across the world.

According to United Nations (UN) estimates, in 2015, the UAE ranked fifth globally in international migrant stock, with 8.095.126 migrants out of a total population of 9.2 million (https://www.macrotrends.net/countries/ARE/uae/immigration-statistics, accessed on 1 February 2024). As of 2024, the UAE's total population has grown to 10.24 million, according to Statista (https://www.statista.com/statistics/297140/uae-total-population/, accessed on 1 February 2024). Notably, the Indian community comprises the largest expatriate group in the UAE, with a population of 3.89 million as of January 2024 (https://www.statista.com/statistics/297140/uae-total-population, accessed on 1 February 2024).

The UAE's open and tolerant society philosophy has also made it an attractive destination for people seeking a cosmopolitan lifestyle. The multicultural environment and the absence of income taxes add to the appeal of living and working in the UAE. As a result, the country has become home to a diverse expatriate community cohabiting and working with the locals.

As highlighted in the preceding paragraphs, the historical examination of migration patterns in the UAE reveals a complex interplay of economic, political, and social factors. Gennaro (2023) research provides a nuanced understanding of the historical trajectories that have shaped the demographic landscape in the UAE, setting the stage for the involvement of religious charities in responding to the evolving needs of migrants. Building on earlier historical analyses, studies like those by Valenta et al. (2020) and Qureshi et al. (2020) provide updated insights into migration trajectories in the UAE. By incorporating recent demographic shifts and geopolitical changes, these works contribute to a nuanced understanding of the contextual factors influencing the engagement of religious charities with migrant populations.

Several notable case studies shed light on the compassionate initiatives of religion-based charitable organizations in the UAE. Baycar (2022) delve into the operations of the UAE Red Crescent and the Mohammed bin Rashid Global Initiatives, offering insights into the multifaceted approaches employed to address the diverse needs of migrant populations. The past few years have witnessed a growing body of literature presenting emerging case studies of compassion. Notably, the research by Kaag and Sahla (2020) delves into the operations of newly established religious charitable organizations, offering insights into innovative approaches and responses to the evolving needs of migrants in the UAE.

The social integration of migrants is a critical aspect of their well-being. Studies by Mathews and Zhang (2017) and Zanfrini (2020) underscore the transformative role of religious charities in promoting social cohesion. By providing not only material support but also fostering cultural understanding and community bonds, these organizations contribute significantly to the integration of migrants into the social fabric of the UAE. Xu and Zheng (2023) and Baumann (2014) continue to explore the impacts of religious charities on the social integration of migrants. These works emphasize these organizations' ongoing efforts to provide immediate relief and facilitate long-term integration, recognizing the importance of cultural understanding and community building. Other studies recognize the roles of these organizations in providing emotional, psychological, economic, and social capital support for migrants (Hynie 2018; Stark and Wang 2002). The commitment of religious organizations to humanitarian causes is deeply rooted in ethical imperatives. Kraft and Jonathan (2019) emphasize the moral obligation that religious institutions feel to alleviate human suffering, providing a foundation for understanding the charitable activities of faith-based organizations in the UAE and their impact on migrant communities. The research works of Miller (2015) and Khafagy (2020) underscore the contemporary ethical imperatives that drive religious organizations to engage in humanitarian endeavors. These studies emphasize the evolving nature of faith-based initiatives, reflecting changing societal expectations and the moral

responsibility perceived by these organizations in addressing the needs of migrants. Not surprisingly, charitable organizations face challenges as they evolve in today's ever-changing societal landscapes.

Lehmann and McLarren (2023) identify legal constraints, resource limitations, and cultural disparities as recurrent challenges religious charities face. Addressing these challenges requires innovative approaches, and further research is needed to explore collaborative frameworks and innovative solutions that enhance the efficacy of faith-based initiatives in supporting migrants. CAF (2022) and Martinez-Damia et al. (2023) have identified and explored new challenges facing religion-based charitable organizations, including the impact of global events and pandemics. These studies emphasize the need for innovative solutions and adaptive strategies to address emerging obstacles and continue to support migrant communities effectively. Doing this is essential as the UAE population is expected to grow to around 15.5 million in 2050. Of this growing population, the diverse immigrant communities are the majority contributing to the economic growth and development of the country along with the citizens (https://www.uae-embassy.org/discover-uae/society/human-rights, accessed on 4 February 2024). Based on these points and the intent articulated in the introduction, we adopted the approach and research questions described in the methods section.

## 3. Materials and Methods

### 3.1. Research Design and Questions

In this study, we employed a qualitative research design, as it allows for a detailed exploration of the lived experiences of small local charities, drawing on the interpretive research paradigm. Qualitative methods are particularly well suited for understanding the nuances and contextual factors contributing to the initiatives initiated, strategies adopted for operation, and challenges these organizations face (Creswell 2013). Essentially, the research questions informing this research approach are as follows:

- Research Question 1: How do the charitable organizations describe their missions and sources of inspiration in supporting immigrants in this research context?
- Research Question 2: How do the charitable organizations describe the primary services they provide and their impacts on immigrants in this research context?
- Research Question 3: How do the organizations describe their strategies for adequately supporting immigrants and how others can complement their efforts in this research context?

### 3.2. Sampling

Using the purposive sampling technique (Patton 2014), we elected to collect data from a small number of religious charitable organizations interested in supporting immigrant residents. To this end, we carefully explored the organization's mission, services offered to immigrants, and success stories in helping immigrants, leading to a comprehensive understanding of their initiatives and modus operandi. Specifically, we selected ten charity organizations in the UAE's northern parts. However, we succeeded in collecting data from five of the targeted entities that eventually consented to provide us with the required data (see Table 1). We opted to analyze the data collected, as our intention was not to generalize but to seek an understanding of how charity organizations support immigrants by drawing on religious philosophy (Aderibigbe 2011). The organizations are established and funded by highly placed people and governments.

**Table 1.** Metadata of the anonymized organizations involved in the study.

| S/N | Organization | Year of Creation | Emirate | Beneficiaries |
|-----|--------------|------------------|---------|---------------|
| 1 | Organization One | 1979 | Dubai | Citizens and residents |
| 2 | Organization Two | 2013 | Umm Al Quwain | Citizens and residents |
| 3 | Organization Three | 1979 | Umm Al Quwain | Citizens and residents |
| 4 | Organization Four | 1989 | Sharjah | Citizens and residents |
| 5 | Organization Five | 1980 | Dubai | Citizens and residents |

*3.3. Data Collection and Analysis Procedures*

In collecting data from the organizations, we developed a questionnaire with eight qualitative questions. We validated the questionnaire elements by collaboratively reviewing and revising the texts, ensuring that participants could easily understand the content and that the study's objectives were addressed. After this, the fourth author approached the selected charity organizations to distribute the questionnaire and returned to collect them as advised by the organizations. We anonymized the charities' names for confidentiality and privacy purposes.

In analyzing the data collected, we adopted the approach described by Aderibigbe et al. (2022), involving an inductive thematic analysis with themes and supporting codes emerging after identifying patterns in the thoughts expressed (Strauss and Corbin 1998). We then presented the findings, showing how the research questions were answered and highlighting the core themes and supporting quote vignettes.

**4. Results**

*4.1. Research Question 1: How Do the Charitable Organizations Describe Their Missions and Sources of Inspiration in Supporting Immigrants in This Research Context?*

We asked the organizations to describe their missions and how they specifically support immigrant communities. One theme emerged from the data analysis.

4.1.1. Help for People in Need

A common explanation for charitable organizations' mission in this study is to assist people in need. Some help people outside the country in addition to providing support for the residents, while others restrict their services to residents of their territory:

> *The organization is a charitable association dedicated to providing financial, material, and infrastructural support both within and outside the country through external entities, both within and outside the state. This support includes initiatives such as digging wells, constructing orphanages, and local projects focused on education and healthcare. (Org. 1)*

> *The organization is a charitable association established in 1979 and provides support to citizens and residents in need within its geographic vicinity. (Org. 3)*

> *This Charitable and Humanitarian organization's main goal is helping the people in need. It provides assistance to all citizens and residents in various areas of the Emirate. We support the residents in paying schools fees, medical expenses, rental issues, and electricity bills. (Org. 2)*

> *The Charity Association was established in 1989 with the aim of carrying out humanitarian and charitable initiatives. Over three decades, the association has consistently provided humanitarian, health, educational, and cultural assistance to developing countries. Additionally, it has offered essential social services, supported orphan sponsorship, and assisted financially challenged families, both citizens and residents. (Org. 4)*

As the data showed, the organizations' primary assistance locally is financial, covering school, rental and utility, and medical expenses. On the other hand, the provision of infrastructure, such as the digging of wells and the construction of orphanages, takes center stage in support provided internationally. Having asked questions about the organization's mission, we wanted to know what inspired the establishment of the organization. Below, we present the central themes emerging from the data analyzed.

4.1.2. Providing Social Solidarity

In terms of the inspiration for establishing the organizations, the data analyzed indicate the provision of social solidarity for people as the main stimulus for the organizations.

> *Contributing to achieving social solidarity by providing financial and in-kind assistance to needy and needy individuals and families within city and adopting charitable programs*

*and projects that support charitable works and bring general benefit to segments of society. (Org. 2)*

*The association was established by the ruler of the emirate in 1979 with the aim of assisting those in need with limited income, individuals with special needs, the elderly, needy patients, and others. (Org. 3)*

*To achieve social and humanitarian solidarity locally and globally by efficiently and effectively providing charitable programs and services. (Org. 5)*

Providing support can also ensure that people do not take drastically regretful steps to reduce the suffering of their family members, such as engaging in crimes.

*In order to safeguard communities from the anticipated negative impacts due to the lack of support for immigrants, such as the spread of crimes and drugs, measures need to be taken. (Org. 1)*

*The overarching goals of the association include engaging in charitable and public welfare activities. (Org. 4)*

*This narrative underscores the importance of societal solidarity, where the ethos of giving and mutual support prevails. In the United Arab Emirates, such acts of generosity contribute to fostering a culture of care and support, creating a society where both citizens and residents can experience stability and happiness. (Org. 4)*

As the results revealed, supporting migrants ensures they are integrated into their new community with fewer challenges and a sense of belonging. It also ensures that society is crime-free, as migrants will not feel tempted to commit crimes when supported. In terms of the inspiration for establishing the organizations, the data analyzed indicate the provision of social solidarity for people as the primary stimulus for the organizations.

### 4.1.3. Founding Father and Emirati Culture

The data analyzed also indicate that the charitable organizations are inspired by the country's founding fathers and the Emirati culture. These sentiments are enunciated as follows:

*The establishment of the association is based on the directions and humanitarian vision of the state, whose foundations were laid by the founder, the late Sheikh Zayed, may he rest in peace, in service to humanity as a whole. Continuing in this pioneering role, the Emirates Charity Association has undertaken the construction and development of its strategy, relying on the primary source of inspiration found in the vision of the United Arab Emirates in the charitable and humanitarian realm. (Org. 5)*

*The Association operates in accordance with the princely decree issued by His Highness Sheikh Sultan bin Mohammed Al Qasimi in 1989. The unwavering and continuous support stems from the values of compassion and generosity deeply rooted in the Emirati culture, as the association remains dedicated to helping those in need and disadvantaged communities. (Org. 4)*

From the data, it is clear that the support provided by the organizations can help improve social cohesion among diverse residents living in the country. Additionally, the gestures can reduce crime rates, making residents more secure and comfortable dwelling in the country.

### 4.2. Research Question 2: How Do the Charitable Organizations Describe the Primary Services They Provide and Their Impacts on Immigrants in This Research Context?

We asked the participants to clarify the range of services their organizations offer to immigrants. The primary theme that emerged is presented below

4.2.1. Food and Bills' Clearance Support

From the data analyzed, the organizations explained that they provide food and assist immigrants in clearing bills of various types. These sentiments are expressed as follows:

*The charity organization provides food assistance, health assistance, educational assistance and housing assistance. (Org. 1)*

*Within the emirate only (Emirate of XXX), providing financial, food, medical, educational support, as well as housing rental assistance. (Org. 3)*

*The charity helps in many areas, solving their housing, educational, and health problems. The charity provides food supplies to residents, distributes seasonal food supplies to eligible groups, and also provides departure tickets for groups that wish to leave the country, but are unable due to financial hardship. (Org. 2)*

The revelation above re-echoed the previous data, showing that core support for immigrants focused on food, health, education, and housing assistance. In providing food support, they also target some seasons to ensure everyone celebrates and feels included in the seasonal celebration moods.

*The charity helps in many areas, solving their housing, educational, and health problems. The charity provides food supplies to residents, distributes seasonal food supplies to eligible groups, and also provides departure tickets for groups that wish to leave the country, but are unable due to financial hardship. (Org. 2)*

*It offers support for families, including sacrificial offerings, iftar meals, food distribution, water supply, and subsidies for Hajj and Umrah. (Org. 5)*

Through the services highlighted, the organizations contended that immigrants receive the required support to lead quality lives in their communities. In our quest to understand the impact of the services and support mentioned, we asked the organizations to share some success stories documenting the positive effects of their efforts with us. With a sense of fulfillment, they indicate that financial support assisted their immigrant beneficiaries in some ways, as shown below.

4.2.2. Financial Support

They indicated that they were able to solve a variety of problems for immigrants within their communities. These sentiments are captured in the following texts:

*Many cases have been supported by the association in terms of school and university fees, and this support has had a significant impact on the graduation of the son with a university degree, which was challenging to attain given the family's circumstances and the low income. (Org. 1)*

*The husband was imprisoned in the rental case, and his children and wife stayed in a rented room without food or drink. The wife wanted to return with her children to her homeland, but because of the expiration of the residency and the presence of fines, her travel was hindered. The Foundation provided assistance by providing food supplies to the family, paying the family's residency fines, and providing tickets so that the family could return to their homeland. (Org. 2)*

Explaining further, another charity organization explains that the financial support it provides includes:

*Sponsorships and humanitarian care, encompassing initiatives such as orphan sponsorship, sponsorship of students of knowledge, sponsorship of Quran teachers, and family sponsorships. (Org. 5)*

Even though the third organization did not mention financial support, it did allude to the view that its efforts were beneficial to the immigrants:

*I don't have a specific case in mind since the process involves a continuous cycle among various employees and departments. The overall success of the aid extended to vulnerable*

*immigrants, whether in healthcare or other realms, has a substantial impact on us. (Org. 3)*

From the data, we can assume that financial support assisted the immigrants in this research setting to solve problems, including flight ticket money, tuition fees, and healthcare bills.

*4.3. Research Question 3: How Do the Organizations Describe Their Strategies for Adequately Supporting Immigrants and How Others Can Complement Their Efforts in This Research Context?*

We asked the organizations to clarify their strategies for effective service delivery to determine the immigrants' needs. From the data analyzed, the central theme that emerged is presented below.

### 4.3.1. Direct Communication

As the data revealed, the residents and immigrants directly walk in or call the organizations by phone to discuss their predicaments and request support:

*Through direct communication with them or by directly requesting from them. (Org. 1)*

*The affected group contacts the organization by calling or attending in person. There are rare cases where we are notified through benefactors or other parties. (Org. 2)*

*Residents seeking assistance can visit the association's office to explain their circumstances. After a thorough evaluation of their situation, the association provides them support. (Org. 3)*

Interestingly, the data indicate that immigrants making requests are supported after thorough examinations of their situations and requests made. Doing this ensures that resources are well deployed by the organizations and appropriately used by immigrants. In addition, we asked the organizations to explain their strategies to ensure cultural sensitivity and inclusivity in serving a diverse immigrant population. The main theme that emerged from the data analyzed is reported as follows.

### 4.3.2. Competent and Culturally Sensitive Approach

We asked the organizations to explain their strategies for ensuring cultural sensitivity and inclusivity while serving immigrants from different backgrounds. According to the organizations, they engage with immigrants by drawing on the skills of staff who are culturally sensitive, open to differences, and equipped with the insight to foster inclusivity:

*Through a qualified staff capable of interacting with diverse nationalities. (Org. 1)*

*Providing financial assistance is not just our sole focus; emotional support is also integral. Engaging with diverse cultures is a positive skill I've acquired through my professional experiences. (Org. 3)*

*The association offers its services to all residents without distinction, irrespective of gender, color, tribe, or any similar factors. (Org. 5)*

While reinforcing the essential place of competency in promoting inclusivity with cultural sensitivity considered, it is also believed that organizations with the right skills and interests can complement their efforts. An organization explains thus:

*In coordination with the competent authorities to provide medical awareness courses and first aid courses and to encourage residents to attend them. (Org. 2)*

As the data showed, cultural competency and collaboration with other agencies assist the organizations in providing adequate support for the immigrants in this study. Thus, we asked the organizations to discuss further how they work with others and clarify how others can complement their efforts. On the one hand, they commonly identified the following theme as how they work with other entities to support immigrants.

### 4.3.3. Collaborative Agreements with Other Agencies

A significant way through which the organizations work with others to support immigrants is through collaborative agreement. This concept is explained as follows:

*The association believes in the importance of cooperation and partnerships in its work, aiming to achieve greater results and accomplishments. Collaborations between relevant entities are a key pillar in the agenda of sustainable development, pursued collectively by all parties involved. (Org. 5)*

*Other charitable organizations or community centers, and their involvement is manifested through participation in seasonal campaigns such as Zakat al-Fitr or sacrificial offerings. Distribution occurs through them to individuals listed in their records. (Org. 1)*

*The Sharjah Charitable Association collaborates with various governmental and private institutions as a strategic partner to ensure the provision of necessary assistance to financially disadvantaged families, both citizens and residents. (Org. 4)*

They further explained that the collaboration may include reducing prices and assisting in supporting by offering coupons that can be redeemed within designated stores:

*The methods vary according to the authorities, as the organization concludes an agreement with the services Provider company or organizations to obtain better services at prices commensurate with both parties. (Org. 2)*

*Food assistance is provided through coupons in collaboration with Lulu, and medical aid, as well as medications, are facilitated through a partnership with Rua Pharmacy. Surgical and medical procedures involve cooperation with government hospitals. The level of collaboration is more extensive with government.*

On the other hand, the organizations indicated that the different organizations and individuals can collaborate and complement their efforts, as shown below.

### 4.3.4. Targeted Financial Donation

In terms of measures required to expand the organizations' operations in supporting immigrants through the involvement of others, they indicate the need for more financial donations:

*Food assistance is provided through coupons in collaboration with Lulu, and medical aid, as well as medications, are facilitated through a partnership with Rua Pharmacy. Surgical and medical procedures involve cooperation with government hospitals. The level of collaboration is more extensive with government. (Org. 3)*

*Providing financial donations from charitable and Zakat funds, as well as food donations, or offering free or nominally priced services. (Org. 1)*

*Support is provided through various channels, including fundraising initiatives, periodic campaigns, individual donations in collection boxes for cash, clothing, books, and recyclables. Additionally, collaboration with a recycling company contributes to receiving cash amounts that further aid the association's efforts. (Org. 3)*

*Additionally, the association conducts several seasonal campaigns, programs, and initiatives in collaboration with external supporting entities to guarantee the delivery of aid to those in need. (Org. 4)*

*Members of the community, government bodies, companies, private institutions, and charitable organizations are invited to actively contribute to supporting the diverse programs, initiatives, and activities of the association, both within and outside the country. (Org. 4)*

As the data showed, they indicate that people can be encouraged to donate to mark some special seasons and values in Islam, such as Zakat, to facilitate the donation collection process. Using stations for donating and partnering with other organizations can also help ensure maximum support for the charity organizations.

## 5. Discussion

In this study, we explored the roles of religious charitable organizations in ensuring migrants feel welcomed, supported, and integrated into communities in the UAE. Such information is essential as the country's population will continue to grow with the immigrant population in the majority (Froilan and Tsourapas 2021; Statista 2023, accessed on 1 February 2024). From the data analyzed, we found that the charitable organizations' missions were grounded in their intention to support people in need and provide social solidarity within the country. The organizations further explained that the visions of the country's founding fathers and Emirati culture also influenced their missions. These organizations' initiatives align with religious principles, assisting in food, shelter, healthcare, education, and legal support (Luke 10: 25–37; Surah Al-Baqarah 2: 267). In addition, these initiatives are driven by the UAE government's philosophy emphasizing the need to provide humanitarian assistance to people in need within and outside the UAE (UAE Foreign Aid 2021 Retrieved from https://www.mofa.gov.ae/en/The-Ministry/UAE-International-Development-Cooperation/Annual-Foreign-Aid-Report, accessed on 10 December 2023). Not surprisingly, these religious organizations often work with the government and other stakeholders, contributing to emergency relief efforts, long-term integration programs, and advocacy for migrants' rights. Thus, collaborative endeavors should be encouraged between charity bodies and other entities, including the government and nongovernment organizations. This is essential for a more comprehensive and inclusive approach to supporting migrants in the UAE. These findings also indicate the need for the younger generation and people in the country to be more encouraged and educated to treasure humanitarian acts and empathy toward others. Not surprisingly, the UAE Vision 2021 plan emphasized the need for everyone in the country to unite in responsibility, destiny, knowledge, and prosperity. This requires teaching religious principles and UAE culture and fostering tolerance, diversity, and compassion toward others regardless of nationality, gender, and religious affiliations.

Drawing on their missions, the charitable organizations' primary services supporting migrants in this study context are food supply, clearing of bills, and other financial support. While they provide food regularly and during festive periods, the organizations constantly help clear the hospital, rent, and school fee bills of migrants. These acts reinforce the humanitarian visions of the country's founding fathers and the UAE society's values grounded in Islamic teachings, promoting the kind acts of sharing and modeling the practice in training the younger generations (Surah Al-Baqarah 2: 267; Hadith: Al-Bukhari). Rendering these services has been reported to be impactful, as many migrant families have benefited from their compassionate and supportive gestures. For instance, university tuition and air ticket fees for indigent migrant families impacted not only the migrants but also the organizations, as indicated by one of the organizations (Org. 3). This implies that being good to others can positively impact the compassionate entities. Thus, inculcating the spirit of supporting others should be championed by all societal stakeholders, including government, schools, nongovernment organizations, and families. Using authentic and real-life examples of individuals with visions for supporting others, such as the efforts and exemplary leadership of the UAE's founding fathers, will ensure the measures' effectiveness.

Regarding the strategies employed in determining the needs and providing the necessary support to migrant families, the organizations embrace an open-door policy. As the data revealed, they encourage migrants to reach out directly by walking in or emailing them to seek support for addressing their plights. This open-door approach aligns with the need for the diverse population in the country to be supported as part of the country's commitment to equality and social justice (https://www.uae-embassy.org/discover-uae/society/human-rights, accessed on 3 February 2024). However, more publicity to ensure that migrants understand the strategies to contact them will be essential. As reported, there is a need for education and advocacy to enhance migrants' social cohesion (U.S. Conference of Catholic Bishops 2021). In ensuring that migrants are addressed or attended to without contempt, the organizations explained that they employ staff with cultural competencies

required in diverse societies. This type of engagement may be influenced by religious teachings and philosophies emphasizing treating everyone with respect and compassion regardless of differences (Aderibigbe et al. 2023; Zanfrini 2020). In addition, supporting migrants will not be effective without collaborative endeavors. Rodríguez-Valls and Torres (2014) highlight the crucial place of partnership in comprehensive support for migrants. Not surprisingly, charitable organizations collaborate with different agencies, including government and private organizations, to foster adequate support for migrants in this context. Thus, efforts must be channeled to promote collaborative endeavors between different entities in the interest of migrant families. To accomplish this effectively, government intervention will be required, and economically successful individuals can also support the organizations through targeted donation drives.

## 6. Conclusions

Drawing on the findings of this study, it is fair to conclude that solidarity towards migrants in the UAE underscores the pivotal role of religious and charitable organizations in fostering a sense of compassion, support, and unity toward migrants. The diverse initiatives these organizations undertake, rooted in various religious principles, contribute significantly to addressing the multifaceted needs of migrant communities in the UAE. The synergy between government and nongovernment actors showcases the power of collaborative endeavors transcending sociocultural and religious boundaries to provide essential aid and promote social integration. Essentially, the study's findings affirm the positive impact of religious charities in promoting the well-being and social cohesion of migrants, exemplifying a shared commitment to humanitarian values and reinforcing faith in the collective capacity for compassion and solidarity. In addition, the findings reinforce the need to enhance the teaching and modeling of values promoting compassion, harmony, diversity, and social cohesion, drawing on Emirati values and Islamic principles. Engaging in this noble act will require the concerted efforts of agents of socialization, including family, schools, religious institutions, and government.

While the study provides insight into the charitable organizations' modus operandi in the UAE and offers room for the transferability of ideas, generalization from the findings is impossible. If logistically realistic, future studies may consider multiple research approaches, including mixed methods and quantitative studies with a bigger sample size. It will also be beneficial to comprehensively appraise their programs' enduring impact, investigate the dynamics and hurdles in interfaith collaborations, and scrutinize these organizations' role in shaping policies related to migrants. Research endeavors should extend to exploring community perceptions of initiatives supporting migrants, encompassing the viewpoints of both migrants and the local populace. Furthermore, scholars are encouraged to delve into inventive approaches within charitable initiatives, undertake comparative studies to discern global trends, and evaluate ethical considerations such as cultural sensitivity and the potential for proselytization. A comprehensive understanding of how religious charities contribute to migrant integration, foster social cohesion, and address gender-specific challenges is imperative. Additionally, researchers should acknowledge the dynamic nature of migration patterns and examine how these organizations adapt to evolving demographics and geopolitical shifts. Ultimately, such research aims to provide insights to guide the development of more effective, inclusive, and ethically grounded policies and practices for supporting migrants in the UAE.

**Author Contributions:** Conceptualization, W.B.H. and S.A.A.; methodology, W.B.H., S.A.A., M.I. and M.M.A.; validation, W.B.H. and S.A.A.; investigation W.B.H. and S.A.A.; Resources, W.B.H., S.A.A., M.I. and M.M.A.; writing—original draft preparation, W.B.H.; writing—review and editing, W.B.H., S.A.A. and M.I. All authors have read and agreed to the published version of the manuscript.

**Funding:** This research received no external funding.

**Institutional Review Board Statement:** This study was conducted in accordance with the Declaration of Helsinki. Data collected were analyzed and reported anonymously. There are no identifiers linked to the charitable organizations or individuals. More so, the study has no potential harmful influences or impacts on individuals or charitable organizations.

**Informed Consent Statement:** Informed consent was obtained from all organizations involved in the study. The qualitative questionnaire's introductory statements clarify the purpose of the study, the expectations from the respondents/organizations, and how the data provided will be used. The questionnaire also indicates that organizations consent to the data provided to be used for this study if the questionnaire is completed.

**Data Availability Statement:** Data are contained within the article.

**Acknowledgments:** We appreciate the charitable organizations and their representatives who gladly shared information on their roles in supporting migrants as religious organizations. Their participation informed the empirical data documented and shared in this study.

**Conflicts of Interest:** The authors declare no conflicts of interest.

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
