# Peer review of "Faith in Humanity: Religious Charitable Organizations Solidarity towards Migrants in the United Arab Emirates"

_religions, doi:10.3390/rel15030266_

Round 1

Reviewer 1 Report

Comments and Suggestions for Authors

In the first instance I would like to acknowledge the potential importance of an article such as this which discusses the activities of religious charities in relation to migrant and related groups in the United Arab Emirates. This is not least because there is, overall, insufficient work that deals with this area. Therefore, overall, this article is to be welcomed. 

In addition, in terms of the what the article has research, reports and discusses concerning the "helping" activities of charities, it provides important source material that can make a distinctive contribution to a broader global discussion of similar phenomena and activities.

However, as the text currently stands, there are two main deficiencies which I believe must be addressed before it would be possible for the article to be published in a peer-reviewed context.

First, the somewhat "laudatory" statements about migrants in the UAE that appear in lines 29-31 and lines 40-43 are, I am afraid, too simplistically asserted as what would seem to be "factual descriptions" in relation to which it might seem as if there were no contestation or problematisation. While the author may in the end reach an evaluative position that is broadly in support of these statements, it would not be appropriate for a peer-reviewed journal article to be published which, despite the labour reforms that have been carried out in recent times, did not also acknowledge, reference, and engage with differing evaluations of the Kafala (or private sponsorship) system which can be found in a number of places, of which only one example would be the Carnegie Foundation's report on "The UAE's System: Harmless or Human Trafficking?" (see https://carnegieendowment.org/2020/07/07/uae-s-kafala-system-harmless-or-human-trafficking-pub-82188). 

Second, in relation to the broader discussion of the activities of religious charities and their engagement with government "powers-that-be", in lines 135-137 of the article, the authors note the example that, in the USA, a range of religious groups have advocated for comprehensive immigration reform ad supported policies that promote the rights of undocumented immigrants. However, unless I have missed something, the article does not contain any examples or discussion of the possibility for such advocacy by religious charities in the context of the UAE.

However, if the author(s) can pay appropriate attention to the above points by engaging with them in a substantive way and with reference to other relevant author discussion, then, overall, the article is well-written and I have no issues with the research methods and data, given the article contains an acknowledgement of the relatively limited nature of the data on which it is based.

Author Response

Thank you very much for taking the time to review this manuscript. I revised the manuscript and took in consideration your valuable comments. I would like to note that I Have reformulated and modified the paragraph including the lines 29-31 and 40-43 according to your comments that is highlighted as recommended by the editorial office. 

Note : kindly you can find the revised part in page 1 lines 29 - 44.

Reviewer 2 Report

Comments and Suggestions for Authors

Faith in Humanity: Religious Charitable Organizations Solidarity Towards Migrants in the United Arab Emirates is discussing the answers to a questionnaire to six charitable organizations. in my opinion, the study is well done.

Author Response

Thank you very much for taking the time to review this manuscript.

And Best Regards

Reviewer 3 Report

Comments and Suggestions for Authors

The article discusses a relevant and underexplored topic. However, the current text has several issues that need to be addressed, listed in order of significance: - A more precise presentation of the migratory impact on the Emirates. To conduct qualitative research on charitable organizations, it is necessary to gather information on the flow of migrants, including when it started, how many migrants there are, where they come from, their legal status, and their living conditions (such as their job market, salaries, and housing).

The author of the research should explain the criteria used to select the group of charitable associations being studied. Additionally, more information should be provided on the six chosen associations, such as their date of foundation, their founder, how they collect funds that are distributed to immigrants, how many migrants received grants per year, and so on. Overall, the article serves as a sociographical description of voluntary action taken in favor of immigrants by non-profit organizations with religious inspiration. While useful, it is not sufficient for an analytical article on the phenomenon. To create a scientifically relevant contribution to knowledge of the socio-religious reality of the United Arab Emirates, the qualitative analysis should be integrated with a reflection on the migratory impact on the Emirates.

Author Response

Thank you very much for taking the time to review this manuscript. I revised the manuscript and took in consideration your valuable comments :

  • I added data related to migration patterns (Page 1 in lines 147-168).
  • I delineated the criteria employed in the selection process of the charitable associations under investigation and furnished details regarding the five selected associations, including their respective dates of foundation, as well as information pertaining to their founders (Page 5 in lines 241-251 and page 6 lines 253-254).

  • The discussion section has been expanded with additional paragraphs inserted on page 12, lines 567-570 and lines 585-587, as well as on page 13, lines 610-613. Furthermore, new references have been incorporated to support the content of the added paragraphs.

all additional data are highlighted as recommended by the editorial office.

Best Regards

Round 2

Reviewer 3 Report

Comments and Suggestions for Authors

The author made revisions to the text based on the remarks and advice that I had suggested.

Author Response

Dear respected Editor,

Thank you very much for taking the time to review this manuscript. WE revised the manuscript and took in consideration your valuable comments :

  • We checked the accuracy of names and affiliations.
  • We added the full address: department, university, city, post code, country of authors.

  •  

    We have carefully checked all author names and we only made correction to the name of Dr. Mesut that has been highlighted. 
  • We revised all references citation and added detailed reference information in the reference list if needed and also as asked by the editor.
  • We have updated the references list, incorporating the necessary modifications and organizing it in alphabetical order.

Best Regards
